# The response of tropical precipitation to Earth's precession: The role of energy fluxes and vertical stability

Chetankumar Jalihal[1,2], Joyce Helena Catharina Bosmans[3,4,a], Jayaraman Srinivasan[2], and Arindam Chakraborty[1,2]

[1]Centre for Atmospheric and Oceanic Sciences, Indian Institute of Science, Bangalore, India
[2]Divecha Centre for Climate Change, Indian Institute of Science, Bangalore, India
[3]Previously at: Faculty of Geosciences, Utrecht University, the Netherlands
[4]Previously at: Royal Netherlands Meteorological Institute (KNMI), the Netherlands
[a]Now at: Department of Environmental Science, Radboud University, the Netherlands

**Correspondence:** Chetankumar Jalihal (jalihal@iisc.ac.in)

**Abstract.**

The changes in Earth's precession have an impact on the tropical precipitation. This has been attributed to the changes in seasonal solar radiation at the top of the atmosphere. The primary mechanism that has been proposed is the change in thermal gradient between the two hemispheres. This may be adequate to understand the zonal mean changes but cannot explain the variations between land and oceans. We have used a simple model of the intertropical convergence zone (ITCZ) to unravel how precipitation changes with precession. Our model attributes the changes in precipitation to the changes in energy fluxes and vertical stability. We include the horizontal advection terms in this model, which were neglected in the earlier studies. The final response of the land and oceans is a result of complex feedbacks triggered by the initial changes in the insolation. We find that the changes in precipitation over the land are mainly driven by changes in insolation, but over the oceans, precipitation changes on account of changes in surface fluxes and vertical stability. Hence insolation can be a trigger for changes in precipitation on orbital timescales, but surface energy and vertical stability play an important role too. The African monsoon intensifies during a precession minimum (higher summer insolation). This intensification is mainly due to the changes in vertical stability. The precipitation over the Bay of Bengal decreases for minimum precession. This is on account of a remote response to the enhanced convective heating to the west of Bay of Bengal. This weakens the surface winds and thus leads to a decrease in the surface latent heat fluxes and hence the precipitation.

## 1 Introduction

The most dominant temporal mode in insolation and tropical precipitation is the 23,000-year precession cycle of the Earth (Berger, 1978; Kutzbach, 1981; Pokras and Mix, 1987). Both proxy (Wang et al., 2007, 2008; Cruz Jr et al., 2005) and model (Kutzbach, 1981; Kutzbach et al., 2008; Tuenter et al., 2005; Weber and Tuenter, 2011; Caley et al., 2014; Shi, 2016) based studies suggest that the intensity of monsoons have varied in proportion to insolation on orbital timescales. When changes in precession increase the insolation in the northern hemisphere, the zonal mean precipitation band shifts northward on account of

the increase in thermal gradient between the two hemispheres (Donohoe et al., 2013; Schneider et al., 2014; Kang et al., 2008). This mechanism cannot explain the longitudinal changes in precipitation (Mohtadi et al., 2016). The simulation of climate models shows that precipitation over land and oceans respond differently to precessional forcing (Clement et al., 2004; Tuenter et al., 2003; Chamales, 2014). This has been observed in the idealized as well as realistic precession experiments with climate

models (e.g., Braconnot et al., 2008; Zhao and Harrison, 2012; Bosmans et al., 2015).

It is attributed to the land–sea contrast theory, in the previous studies (Zhao and Harrison, 2012; Bosmans et al., 2012). The land warms more than the surrounding ocean due to its lower thermal inertia. Hence a low pressure develops over land and a monsoon circulation is established. The increase in insolation leads to deeper thermal lows over land, which enhance the onshore flow of moisture-laden winds. This leads to stronger ascent over land and an increase in precipitation. This thermal

contrast, however, disappears after the onset of monsoon due to the cooling of land by precipitation and cloud cover. In fact, in good monsoon years, the land surface temperature is lower (Gadgil, 2018).

Some studies have used the changes in energy balance to understand the response of precipitation to precession (Braconnot et al., 2008; Hsu et al., 2010; Merlis et al., 2013; Chamales, 2014; Battisti et al., 2014). Braconnot et al. (2008) suggested that the net energy in the atmosphere over land and adjacent oceans changes due to precession. The atmosphere then acts to

15 redistribute the excess energy, thereby setting up a land–ocean difference in precipitation. Hsu et al. (2010) showed that the precipitation changes due to precession are related to the changes in the total column energy, which drives changes in vertical velocity. Chamales (2014) on the other hand has argued that the stability over oceans changes, whereas land regions respond by transporting the excess moist static energy. These are, however, generalizations for the entire tropics. The role of local processes and feedbacks might be important in driving regional changes in precipitation. Thus, individual regions need to be

studied separately, to understand the cause of the changes. For example, Battisti et al. (2014) suggested that higher summer insolation leads to a migration of the near-surface moist static energy from the Bay of Bengal to India, before the onset of monsoon. They argued that hence the precipitation centroid shifts to India.

Moisture and MSE equations can be used separately to understand the dynamics of monsoon under different climate scenarios (Sun et al., 2016, 2018). In this work, we follow Neelin and Held (1987) and demonstrate the advantage of combining the

25 two equations. The resulting simple model attributes precipitation to energy fluxes and vertical stability of the atmosphere. This model, however, can only be used for regions where moisture and temperature gradients are weak. In this paper, we propose a modified version of the simple model which takes into account the horizontal gradients as well. We have used time-slice experiments in a high-resolution general circulation model (GCM) EC-Earth (Bosmans et al., 2015). This GCM was run in two orbital configurations which correspond to the extremes in precession (Fig. 1). The advantage of doing this is that the amplitude

of the response is large, while the spatial pattern is similar to a simulation of realistic precession such as Mid-Holocene (MH).

The paper is organized as follows. The next section describes the model, the experimental setup, and outlines the derivation of a simple model for the intertropical convergence zone (ITCZ). We have used this simple ITCZ model to understand the factors leading to the shift in precipitation between land and oceans, at the regional scale. The results are described in section 3. It is followed by a discussion about the precipitation response to MH and obliquity forcing with the help of the simple ITCZ

model.

## 2 Experimental design and analysis method

### 2.1 Climate model description

EC-Earth is a fully coupled ocean-atmosphere GCM (Hazeleger et al., 2010, 2012). We have used the model version 2.2. The Integrated Forecasting System (IFS) was the atmospheric component. The spectral resolution was T159 (roughly 1.125° x 1.125°) with 62 vertical levels. The convective scheme Bechtold et al. (2008) was used along with the Balsamo et al. (2009) land surface scheme H-TESSEL, including surface runoff. The Nucleus for European Modeling of the Ocean (NEMO, version 2) was the ocean component. The horizontal resolution was 1° with 42 vertical levels (Madec, 2008; Sterl et al., 2012). NEMO includes sea-ice model LIM2. OASIS3 coupler (Valcke and Morel, 2006) couples the ocean, sea-ice, land, and atmosphere. EC-Earth performs well for the present day when compared to CMIP3 models in terms of climatology as well as inter-annual, spatial and temporal variability (Hazeleger et al., 2010, 2012).

### 2.2 Experimental designs

The two precession extremes, Precession minima: $P_{min}$ and Precession maxima: $P_{max}$ correspond to summer solstice at perihelion and winter solstice at perihelion respectively (Fig. 1). Table 1 shows the orbital configurations used. This leads to a stronger seasonal cycle in the Northern Hemisphere (NH) and a weaker seasonal cycle in Southern Hemisphere (SH) in $P_{min}$ (Fig. 2). On the other hand, the seasonal cycle is weaker in the NH and stronger in the SH in $P_{max}$.

The model is run separately for each of the orbital configurations. The length of each simulation is 100 years, with the first 50 years being considered as spin-up. We have used the climatology of the last 50 years, for all our analysis. The orbital parameters remain constant throughout the simulation. All other boundary conditions (e.g. the solar constant, greenhouse gas concentrations, orography, ice sheets, vegetation) were kept constant at the pre-industrial levels. Vernal equinox has been fixed at 21$^{st}$ March, and the present–day calendar is used. Since the length of the season and the dates of equinoxes change along the precession cycle, the autumn equinoxes do not coincide. This is known as the "Calendar Effect". It introduces some errors due to the phasing of insolation. We do not make any corrections in order to be consistent with previous studies. Further details about the experiments are provided in Bosmans et al. (2015).

### 2.3 Diagnostic methodology

Hadley cell is a thermally direct overturning circulation in the tropics. It takes energy away from the tropics and transports it towards the poles. The Hadley cell has a rising branch in the deep tropics and a descending branch in the extra-tropics. This leads to moisture convergence near the rising branch. ITCZ coincides with the rising branch of the Hadley cell and is responsible for the zone of heaviest precipitation in the tropics. The characteristics of the ITCZ can be described by using the conservation equations for Moist Static Energy (MSE) and moisture. Using this approach, Neelin and Held (1987) proposed a simple model for ITCZ in terms of net energy input into the atmosphere and vertical stability. This is a diagnostic model, that has been used to explain variations in rainfall due to global warming (Chou and Neelin, 2004; Chou et al., 2006) and

the impacts of aerosols (Chou et al., 2005). In this section, we have discussed this simple model in detail. The Eqs. 1 and 2 correspond to the conservation of MSE and moisture in a vertical column of the atmosphere. The first term in both the equations is horizontal divergence, with the second term being the vertical divergence of MSE and moisture fluxes respectively.

The quantities on the right–hand side are the sum of all sources and sinks. Further details on the derivation of Eq. 1 can be found in Neelin and Held (1987). The time derivatives have been dropped in these equations because the climate is assumed to be in a steady state. The angle brackets $(<>)$ indicate vertical integral.

$$\langle \nabla \cdot m\boldsymbol{U} \rangle + \left\langle \frac{\partial m\omega}{\partial p} \right\rangle = Q_{div} \tag{1}$$

$$\langle \nabla \cdot q\boldsymbol{U} \rangle + \left\langle \frac{\partial q\omega}{\partial p} \right\rangle = E - P \tag{2}$$

$$\langle A \rangle = - \int_{P_b}^{P_t} A \frac{dp}{g} \tag{3}$$

Where,

$P$ - Precipitation rate (mm day$^{-1}$)

$E$ - Evaporation rate (mm day$^{-1}$)

$Q_{div}$ - Total Column Energy, i.e. the sum of all the energy fluxes into the atmosphere (Eq. 4). (in mm day$^{-1}$; taking the latent heat of vaporization as $2.26 \times 10^6$ J kg$^{-1}$ we get 1 mm day$^{-1}$ = 26.16 W m$^{-2}$). Over land since the storage term is small, the sum of all the energy fluxes at the surface is small. Hence, $Q_{div}$ is mainly governed by the fluxes at the Top of Atmosphere (TOA). However, over oceans the contribution of surface fluxes is large.

$q$ - specific humidity (kg kg$^{-1}$)

$m$ – Moist Static Energy (J kg$^{-1}$), which is the sum of internal energy, potential energy and moist energy ($C_pT + gZ + L_vq$)

$P_b$ – Pressure at the bottom of the atmospheric column (Pa)

$P_t$ – Pressure at the top of the atmospheric column (Pa)

$g$ – acceleration due to gravity (m s$^{-2}$)

The full equation for $Q_{div}$ is:

$$Q_{div} = \underbrace{LHF + SHF + Net\_Sfc\_Rad}_{\text{bottom fluxes}} + \underbrace{Net\_TOA\_LW + Net\_TOA\_SW}_{\text{TOA Fluxes}} \tag{4}$$

Where,

$LHF$ : Surface Latent Heat Flux (mm day$^{-1}$)

$SHF$ : Surface Sensible Heat Flux (mm day$^{-1}$)

$Net\_Sfc\_Rad$ : Net Surface Radiation (Long wave + Short wave) (mm day$^{-1}$)

$Net\_TOA\_LW$ : Net Top Of Atmosphere Long wave radiation (mm day$^{-1}$)

$Net\_TOA\_SW$ : Net Top Of Atmosphere Short wave radiation (mm day$^{-1}$)

Clubbing all the radiation fluxes together into one quantity '$Q_{rad}$', we get:

$$Q_{div} = LHF + SHF + Q_{rad} \tag{5}$$

Assuming $\omega = 0$ at the top as well as the surface, leaves us with the horizontal terms only. The governing equations can be combined and simplified as:

10 $$P - E = \frac{Q_{div}}{GMS} \tag{6}$$

$$GMS = \frac{m_1 - m_2}{L_v(q_2 - q_1)} \tag{7}$$

$$m_1 = \frac{\int_{P_t}^{P_m} m\nabla \cdot \boldsymbol{U} dp/g}{\int_{P_t}^{P_m} \nabla \cdot \boldsymbol{U} dp/g} \tag{8}$$

$$m_2 = \frac{\int_{P_m}^{P_b} m\nabla \cdot \boldsymbol{U} dp/g}{\int_{P_m}^{P_b} \nabla \cdot \boldsymbol{U} dp/g} \tag{9}$$

Where, GMS is the gross moist stability, as obtained by taking the ratio of the Eqs. 2.11 and 2.12 from (Neelin and Held,

15 1987). $m_1$ and $m_2$ are respectively, the total MSE in the upper (mid-troposphere to top) and lower troposphere (surface to mid-troposphere), normalized by the divergence of that layer. Thus, GMS is mainly a function of vertical profiles of MSE and it provides a measure of vertical stratification of the atmosphere. $P_m$ is pressure at the mid-troposphere level. Similarly, $q_1$ and $q_2$ represent the total moisture in the upper and lower troposphere, normalized by divergence. The mass convergence in the

lower troposphere is the same as the mass divergence in the upper troposphere. The horizontal variations of temperature and moisture are assumed to be weak within the tropics. This implies that the horizontal advection of temperature and moisture are small. This simple model attributes the changes in $P - E$ to either the changes in total column energy or the vertical stability.

Figure 3a shows a scatter of $P - E$ as a function of $Q_{div}$ for the three summer months June, July, and August taken separately. The scatter is made for Central India, the Bay of Bengal and North Africa for each of the precession extremes. We chose these three regions to highlight that neglecting the role of horizontal advection, may not always be appropriate. The plot shows that the two are nearly linear, as indicated by the simple model (Eq. 6). The slight deviations from linearity are due to variations in GMS. As we go from $P_{max}$ to $P_{min}$ (low to high insolation in NH summer), both $Q_{div}$ and $P - E$ increase over Central India and North Africa (land regions). However, both these quantities decrease over the Bay of Bengal (oceanic region). The net energy input into the atmosphere and thus, $Q_{div}$ is positive for all these regions during the summer.

We have shown in figure 3b, a scatter of $P - E$ vs GMS for the same regions. The figure shows that there is no definite relation between the two. Equation 6 suggests that all values for GMS should be positive since $P - E$ and $Q_{div}$ are both positive. There are, however, some points in the scatter where GMS is negative. This indicates that the assumption about the horizontal advection being small is not always valid. Hence, we modify the definition of GMS to include the horizontal advection terms.

By taking the ratio of Eqs. 1 and 2, after multiplying Eq. 2 by $L_v$ (the latent heat of vaporization for water), we get:

$$P - E = \frac{Q_{div}}{TGMS} \tag{10}$$

$$TGMS = \frac{\left\langle \nabla \cdot m\boldsymbol{U} + \frac{\partial m\omega}{\partial p} \right\rangle}{-L_v \left\langle \nabla \cdot q\boldsymbol{U} + \frac{\partial q\omega}{\partial p} \right\rangle} \tag{11}$$

Where, TGMS stands for total GMS (the term "Total" indicates inclusion of all advection terms). TGMS is based on only one assumption, that the time derivatives of $m$ and $q$ are negligible. This is a good assumption for seasonal mean conditions. TGMS is particularly useful for smaller regions, where horizontal advection can be large. TGMS represents how efficiently an atmospheric column can diverge MSE per unit moisture converged into the column. TGMS is an extension of the concept of GMS, with horizontal advection terms included. This suggests that, along with the energy fluxes and vertical stratification of a column, the lateral transport of MSE and moisture determine the precipitation. A value of TGMS similar in magnitude to GMS indicates that the horizontal transport of MSE is negligible. A change in TGMS between two climates would suggest that the transport of MSE has changed. We have used the equivalence in Eq. 1 and 2 to estimate TGMS. Since our goal is not to estimate the changes in $P - E$ but to diagnose the cause of these changes, there is no need to make an independent estimate of TGMS.

To quantify the relative contribution of $Q_{div}$ and TGMS to the changes in $P - E$, we do the following. Writing Eq. 10 for $P_{max}$:

$$P - E = \frac{Q}{G} \tag{12}$$

Where $P$, $E$, $Q$, and $G$ are precipitation, evaporation, $Q_{div}$, and TGMS respectively. Considering $P_{max}$ as the reference case and $P_{min}$ as the perturbed case, we can write the following for $P_{min}$:

$$(P + \Delta P) - (E + \Delta E) = \frac{Q + \Delta Q}{G + \Delta G} \tag{13}$$

Where $\Delta$ represents the perturbation from $P_{max}$. Now dividing by $P - E$, we get

$$1 + \frac{\Delta(P-E)}{(P-E)} = \frac{1 + \Delta Q/Q}{1 + \Delta G/G} \tag{14}$$

This equation can further be modified as:

$$\underbrace{\Delta(P-E)}_{\text{Change in P-E}} = \underbrace{\frac{\frac{\Delta Q}{Q}}{1 + \frac{\Delta G}{G}}(P-E)}_{\text{Contribution from } Q_{div}} + \underbrace{\frac{-\frac{\Delta G}{G}}{1 + \frac{\Delta G}{G}}(P-E)}_{\text{Contribution from TGMS}} \tag{15}$$

## 3 Results

In this section, we have explained the changes in $P - E$ between $P_{min}$ and $P_{max}$ in terms of $Q_{div}$ and TGMS. We start by giving an overview of the entire tropics and then we look at the South Asian monsoon in detail.

### 3.1 Response of tropical precipitation to precession

#### 3.1.1 Spatial patterns of the response

Figure 4 shows the difference in precipitation between $P_{min}$ and $P_{max}$ averaged over the tropical land and oceans separately. Precipitation change over the tropical land is out of phase with the changes in precipitation over the oceans. Amplitude of the change is higher over land than over oceans. Furthermore, over land, the change is of a higher magnitude during the boreal summer than the austral summer. This implies that the northern hemisphere monsoons are more sensitive to precession than the southern hemisphere monsoons. The vernal equinoxes during $P_{min}$ and $P_{max}$ occur on the 21$^{st}$ of March. Therefore, the difference in insolation between the two cases is very small during March. Hence the changes in land and ocean precipitation have a zero crossing during this month. Since we are interested in regions where there is moisture convergence, our analysis will focus on $P - E$ instead of precipitation.

In figure 5, the spatial pattern of the changes in $P - E$ and $Q_{div}$ are shown averaged over JJA (left panel) and DJF (right panel). First, we discuss the changes in precipitation during JJA. Most of the land regions in the northern hemisphere show an increase in $P - E$. The African monsoon has strengthened substantially in $P_{min}$ with an increase of about 10 mm day$^{-1}$ . $P - E$ has, in general, decreased over the oceans. However, there are many regions over the oceans (e.g., the Arabian Sea) where $P - E$ has increased. Hence, the amplitude of the changes in $P - E$ is small when averaged over all the tropical oceans (Fig. 4). The changes in $Q_{div}$ have a pattern similar to that of $P - E$, with positive values over most of the land regions, and both

positive and negative values over the oceanic regions. This is due to the direct relation between $P-E$ and $Q_{div}$ as suggested by the simple model. There are, however, some exceptions like the Arabian Sea and Africa. $Q_{div}$ has decreased over the Arabian Sea but, $P-E$ has increased. The region of the largest increase in $P-E$ and $Q_{div}$ are not co-located over Africa. These are on account of the changes in TGMS.

During DJF, $P_{min}$ has lesser insolation (Fig. 2) and correspondingly a decrease in $P-E$ and $Q_{div}$ is seen over the land regions (Fig. 5c and d). Over oceans, there are regions of both positive and negative changes in $P-E$ during DJF as well. The magnitude of changes in $Q_{div}$ is of a similar order during JJA and DJF. However, the changes in $P-E$ are larger during JJA compared to DJF.

### 3.1.2 Dominant factors determining the response of tropics

In this section, we look at the various terms in Eq. 15 for different regions of the tropics (Fig. 6). The top and bottom panels are for the northern and southern hemispheres respectively. The analysis was done for the summer months of the respective hemispheres (JJA for the northern and DJF for the southern hemisphere). The blue bar represents the changes in $P-E$, whereas the light red and dark red bars are contributions from $Q_{div}$ and TGMS. $Q_{div}$ explains most of the changes in $P-E$ when all the land regions in the northern tropics are taken together. This need not be true in smaller regions. For example, TGMS contributes most to the changes in $P-E$ over Africa. Because $P-E$ has a different sign over various oceanic regions, the change in $P-E$, averaged over all the tropical oceans is small. The contributions from $Q_{div}$ and TGMS are in opposite directions, thus almost canceling each other out. The contribution from TGMS is, however, slightly higher. The Arabian Sea shows an increase in $P-E$, due to a change in TGMS. The decrease in $P-E$ over the Bay of Bengal is, however, mainly due to changes in $Q_{div}$ with the changes in TGMS being small.

In the southern tropics the dominant contribution is from changes in $Q_{div}$ over land and changes in TGMS over oceans. In the case of South Africa and Brazil changes in TGMS and $Q_{div}$ make an equal contribution. TGMS drives most of the changes in $P-E$ over northern Australia and South Atlantic. Figure 6 highlights the fact that the mechanisms for the changes in precipitation are region specific. Hence, each region must be studied separately to understand the physical mechanism that caused the changes in $P-E$. Both the Indian land mass and the Bay of Bengal are part of the Indian monsoon system, yet they demonstrate a different response to the precessional forcing. Hence, we discuss this asymmetric response in detail in the following subsection. Such an asymmetry also exists within the East Asian monsoon, which has been discussed in a separate subsection.

### 3.2 Explaining the response of the Indian monsoon to precession

Battisti et al. (2014) suggested that different response of the Indian land mass and the Bay of Bengal is due to migration of near-surface equivalent potential temperature from the Bay of Bengal over to India. This is because the rate of increase in insolation is higher in the high insolation (similar to $P_{min}$) experiment. This causes the equivalent potential temperature $\theta_e$ to rise rapidly over India. It is known that the location of ITCZ coincides with that of the surface energy maxima (Privé and Plumb, 2007a, b; Bordoni and Schneider, 2008; Boos and Kuang, 2010). Hence ITCZ migrates over India quickly and remains

there. However, EC-Earth simulates higher near-surface equivalent potential temperature $\theta_e$ over the Bay of Bengal, in both $P_{min}$ and $P_{max}$ (Fig. 7). In this section, we propose an alternate mechanism for the different response of the Indian land mass and the Bay of Bengal to the changes in precession.

We have shown earlier that in $P_{min}$ there is an increase in $Q_{div}$ over the Indian land mass and a decrease over the Bay of Bengal with respect to $P_{max}$ (Fig. 3 and 6). Here we examine the factors that caused the changes in $Q_{div}$. Splitting $Q_{div}$ into its component fluxes (Eq. 5) will help us to determine which flux contributed the most. Figure 8 is a spatial map of the differences in $P-E$, $Q_{div}$ and its component fluxes $Q_{rad}$, LHF, and SHF. $Q_{div}$ has a good spatial coherence with $P-E$, over most of the regions except the Arabian Sea. As was discussed earlier, this is due to the changes in TGMS, which is able to counter the effect of reduced $Q_{div}$. $P-E$ has decreased along the southern parts of the Western Ghats but has increased in the northern parts of the Western Ghats. $Q_{rad}$ bears a resemblance to $P-E$. This suggests that radiative feedbacks from clouds are present. Changes in $Q_{rad}$ are not large enough to counter the decrease in Latent Heat Flux (LHF) over the Arabian Sea and the Bay of Bengal. Thus, the decrease in LHF over these regions reduces $Q_{div}$ there. In fact, $Q_{div}$ and LHF have similar spatial patterns over the oceanic regions. The changes in Sensible Heat Flux (SHF) are small in most places.

We take two regions: one over central India and the other over the Bay of Bengal, to identify the flux which contributes most to the changes in $Q_{div}$. These regions are outlined with black boxes in Figure 8a. The changes in the three components of $Q_{div}$ over these two regions are depicted in the bar chart (Fig. 8f). It shows the dominance of the radiative terms over India, and LHF over the Bay of Bengal, respectively.

LHF is a function of surface wind speed, Sea Surface Temperature (SST) and near-surface relative humidity. LHF increases with an increase in SST and wind speed. SST has increased over the Bay of Bengal and Southern Arabian Sea by about 2°C (Supplementary figure S1). Thus, it cannot explain the decrease in LHF. Hence, we look at the changes in wind speed (Fig. 9). The top panel of Fig. 9 (a and b) show the mean winds at 850 hPa. The shading indicates wind speed and the streamlines show the direction of flow. The axis of the low–level jet (LLJ) has shifted to the north and this has led to a decrease in winds over the Bay of Bengal. Due to LLJ, deep oceanic water upwells along the coast of Somalia. This leads to cooler SSTs over the western parts of the Arabian Sea. Since in $P_{min}$, LLJ has shifted further north, the region of upwelling also shifts north. Thus, leading to cooler SSTs in the west coast of northern Arabian Sea (Supplementary figure S1). Hence, LHF over the Arabian Sea decreases due to weaker winds in the southern parts and colder SST in the northern parts.

The shift in LLJ leads to lesser moisture flux along the southern part of the Western Ghats. Hence, $P-E$ decreases there. At the same time, the LLJ brings more moisture into the northern parts of the Western Ghats, leading to increase in $P-E$. The shift of the LLJ can be seen more clearly in Figure 9c, where the difference in winds between $P_{min}$ and $P_{max}$ is shown. Along the equator, there exists an anomalous low–level easterly over the Indian Ocean. This meets an anomalous westerly from over the equatorial Africa, at around 40°E. This indicates low–level convergence. Furthermore, on the same meridian, there exists a cyclonic circulation to the north (over the Middle-East) and an anti-cyclonic circulation to the south (over Madagascar). This resembles the response of the winds to the heating of an atmospheric column as shown by Gill (1980).

Gill (1980) proposed a simple shallow water model on an equatorial $\beta$ plane to elucidate the role of latent heating on surface winds. In order to represent convective heating due to latent heat release, he introduced mass divergence in the atmospheric

column. When this model was forced with "heating" over a region at the equator and another region to the north of the equator, it produced a Kelvin wave and a mixed Rossby-gravity wave. The Kelvin wave leads to an anomalous low–level easterly and an anomalous low–level westerly along the equator. The easterly is to the east of the heat source and the westerly to the west of the heat source. These anomalous winds thus lead to low–level convergence at the equator, near the region of the heat source.

The mixed Rossby-gravity wave has a cyclonic circulation to the north of the equator and an anti-cyclonic circulation to the south of the equator. The wind response of EC-Earth hence, suggests that the wind patterns over the Indian subcontinent, are driven by atmospheric heating near the equator and off–equator. Examining figure 5a shows that the heat sources correspond to convective heating of the column due to increased precipitation over the West Equatorial Indian Ocean (WEIO) and over the Middle-East (particularly the Red Sea). There are, however, some important differences between EC-Earth and the Gill model.

EC-Earth is a full GCM with non-zero mean background winds, whereas Gill model is linearized with respect to zero mean background winds. Thus, the EC-Earth's response includes non-linear terms as well.

To summarize, the decrease in $Q_{div}$ over the Bay of Bengal is due to lower wind speeds. The winds decrease because of convective heating over west equatorial Indian Ocean and the Red Sea. The convection over the Red Sea is an extension of the African monsoon. Hence, we examine the factors which lead to an increase in precipitation over these regions. The prevailing

conditions in the pre-monsoon month of May, leads to enhanced convection over these regions, later in the summer. Figure 10a and b, show the difference in $Q_{div}$ and $P - E$ for the month of May. Figure 10b shows changes in $P - E$ in shading and the streamlines represent the changes in the wind direction. $Q_{div}$ is higher over Africa, and this causes early onset of the African monsoon (Fig. 10b) and changes the low–level winds along the eastern coast of Africa. The SST along the eastern coast depends on the coastal upwelling. The changes in winds thus reduce upwelling and increase SST. This enhances convection

over the west equatorial Indian Ocean, further leading to low–level convergence. This positive feedback is responsible for the convective heating that persists through the summer months. As the season advances from May onwards, the African monsoon propagates northward. The region of convection over the eastern side of Africa moves over to the Red Sea. This becomes the off-equatorial heat source.

### 3.3 Factors determining the response of South East Asian monsoon to precession

Shi et al. (2012) showed that the South East Asian monsoon and the North East Asian monsoon are out of phase owing to the El-Nino like SST pattern in P$_{min}$. Here we are addressing the differences in the precipitation changes over South East Asia (land) and the adjacent ocean. The domain for South East Asia is shown in figure 8b. Based on the analysis using Eq. 15, we find that the increase (decrease) in $P - E$ over the land (ocean) grids is mainly due to the increase (decrease) in $Q_{div}$ (Fig. 6). Even though $Q_{div}$ is dominant, the contribution of TGMS is higher over the SE Asia (oceanic regions) when compared to

the Bay of Bengal. Once again decomposing $Q_{div}$ into its component fluxes suggests a similar mechanism that leads to the India–Bay of Bengal redistribution of precipitaion (Fig. 8f). The increase in insolation leads to an increase in $Q_{div}$ over the SE Asian land, whereas a decrease in LHF over the oceanic regions leads to a decrease in $Q_{div}$. The convective heating over WEIO and the Red Sea leads to reduced winds, and hence decreased LHF in the North West Pacific.

## 4 Discussion

In this section, we have discussed the similarities between the sets of idealized experiments ($P_{min}$, $P_{max}$) vs (Mid-Holocene (MH), Pre-Industrial (PI)). The MH and PI experiments were conducted with the same model EC-Earth, the details of which are available in Bosmans et al. (2012). The difference in solar forcing between MH and PI is similar to that between $P_{min}$ and $P_{max}$, albeit with a smaller amplitude (Supplementary figure S2). Moreover, MH has an obliquity $0.66°$ higher than PI, and hence it contributes little to the total forcing (Supplementary figure S2b). Previous research with models has shown that the climate response to precession is independent of obliquity (Tuenter et al., 2003). The climate of MH is therefore mainly driven by precession. The peak in the insolation difference between MH and PI is delayed by a month with respect to the insolation difference between $P_{min}$ and $P_{max}$. Hence the largest precipitation changes in MH occur about a month later than in $P_{min}$ (Fig. 4 and supplementary figure S3). Therefore, we consider Jul–Aug–Sep averages for MH. The land–ocean shift in precipitation in MH is qualitatively explained by changes in $Q_{div}$ (supplementary figure S4). Particularly, the displacement of precipitation from the Bay of Bengal to India is due to the same mechanism that drives these changes in $P_{min}$ (Supplementary figures S5, S6, and S7). The SE Asian monsoon also exhibits a land–ocean shift in rainfall. This is due to radiative heating over land as well as the ocean. This suggests that the cloud radiative feedbacks are stronger for the SE Asian monsoon. The changes in LHF are, however, due to the same reason as in $P_{min}$. We also performed the analysis for a set of obliquity experiments $T_{max}$ and $T_{min}$, corresponding to the maximum and minimum tilt, with eccentricity set to zero (Bosmans et al., 2015). The tropical precipitation shows a land–ocean shift in precipitation, but the amplitude of change is small compared to the precession experiments (Supplementary figure S9). The mechanisms leading to this shift are different for obliquity and precession (Supplementary figures S10 and S11).

Models with different levels of complexities: QTCM (Hsu et al., 2010), Quasi-geostrophic model EC-Bilt (Tuenter et al., 2003), GCM with slab ocean (Battisti et al., 2014) and finally the fully coupled model EC-Earth (Bosmans et al., 2018) have all shown a shift in precipitation between land and ocean, when subjected to the precessional forcing. However, there are no proxies for precipitation over oceans to verify this. Since the climate over islands is influenced by the surrounding oceans, proxies obtained from islands can be thought of as a representation of climate over the surrounding ocean. A speleothem chronology from the Baratang cave in the Andaman Islands (Laskar et al., 2013) in this regard, represents precipitation over the Bay of Bengal. This chronology goes back to 4,000 years before present and shows a long-term decreasing trend in precipitation as we move back in time. The time period corresponding to 4 ka being closer to MH has higher summer insolation and proxies over Indian continent register an increase in precipitation (Ramesh, 2001; Patnaik et al., 2012; Zhang et al., 2016; Kathayat et al., 2017). This suggests that the GCMs and observations indicate the response of Indian land mass is different from the response in the Bay of Bengal.

## 5 Summary and conclusions

Using a simple model for ITCZ, we have interpreted the response of a high resolution fully coupled model EC-Earth to precession. The changes in precipitation can be attributed to either the changes in total energy fluxes going into the column

($Q_{div}$) or the changes in vertical stability of the atmosphere (TGMS). We have included the horizontal advection terms in the calculation of TGMS, which were originally assumed to be small (Neelin and Held, 1987). This allows us to use the simple ITCZ model for relatively smaller domains, where horizontal advection terms can be large. TGMS represents the total transport of the MSE. In places where the horizontal transport is weak, TGMS is the same as GMS. Changes in precession

provide an initial forcing. The final response of the precipitation is due to this initial forcing and the consequent feedbacks. These feedbacks are in the form changes in surface energy fluxes and changes in stability of the atmosphere. In agreement with Chamales (2014), we find that precipitation changes between precession extremes over the whole tropics are, due to changes in $Q_{div}$ over land and due to TGMS over the ocean. This generalization is, however, not valid for smaller regions. Within the domain of the South Asian monsoon, insolation drives changes in $Q_{div}$ over the land, whereas latent heat fluxes contribute

most over the oceans. Particularly, the decrease in LHF over the Bay of Bengal and the North–West Pacific is associated with the weakening of the low–level westerlies over these regions. These changes in westerlies are driven by convective heating of the atmospheric column over the western equatorial Indian Ocean and the Middle-east. There are, however, regions where the changes in TGMS is the main cause of the changes in precipitation (e.g., Africa and Arabian sea). We have demonstrated that the simple ITCZ model can be used to explain the precipitation response for any orbital configuration (e.g., MH, maximum

and minimum obliquity experiments).

*Author contributions.* C. Jalihal, J. Srinivasan and A. Chakraborty analysed and interpreted the GCM output. J.H.C Bosmans designed and ran the experiments. C. Jalihal wrote the manuscript with input from all authors. All authors reviewed the manuscript.

*Competing interests.* The authors declare that they have no conflict of interest.

*Acknowledgements.* We thank A. Nikumbh for useful comments. The authors acknowledge support from the Centre for Excellence in the

Divecha Centre for Climate Change (DCCC). This work was partially funded by DST India.

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

**Table 1.** The orbital configuration used for the extremes in precession, Precession minima: $P_{min}$, Precession maxima: $P_{max}$ and the pre-industrial. 'e' represents eccentricity, $\delta$ is the tilt and $\omega$ is the longitude of perihelion. The values of these have been chosen, based on the extremes in the precession parameter $e*sin(\pi+\omega)$, that have occurred in the last 1 Myrs Berger (1978). Pre-industrial values are shown for comparison.

|                    | Eccentricity, $\mathbf{e}$ | Obliquity, $\boldsymbol{\delta}$ (°) | Longitude of perihelion, $\boldsymbol{\omega}$ (°) |
|--------------------|-------------|--------------|-----------------------------|
| **Pre-Industrial** | 0.017       | 23.45        | 282.04                      |
| $\mathbf{P_{min}}$ | 0.056       | 22.08        | 95.96                       |
| $\mathbf{P_{max}}$ | 0.058       | 22.08        | 273.5                       |

**Table 2.** The regions used in this article and their coordinates.

| Region | Co-ordinates |
| --- | --- |
| **Northern tropics** | ( $0°$N – $30°$N;   $0°$E – $360°$E) |
| **Southern tropics** | ($30°$S –  $0°$N;   $0°$E – $360°$E) |
| **Central India** | ($15°$N – $25°$N;  $73°$E – $83°$E) |
| **Bay of Bengal** | ($10°$N – $20°$N;  $85°$E – $95°$E) |
| **South East Asia** | ( $0°$N – $25°$N; $100°$E – $125°$E) |
| **Arabian Sea** | ($10°$N – $20°$N;  $60°$E – $70°$E) |
| **N. Africa** | ( $5°$N – $15°$N;  $20°$W – $0°$E) |
| **Brazil** | ($20°$S – $10°$S;  $70°$W – $50°$W) |
| **South Atlantic** | ($20°$S – $10°$S;  $30°$W – $0°$E) |
| **South Africa** | ($20°$S – $10°$S;  $15°$E – $35°$E) |
| **North Australia** | ($25°$S – $15°$S; $130°$E – $140°$E) |

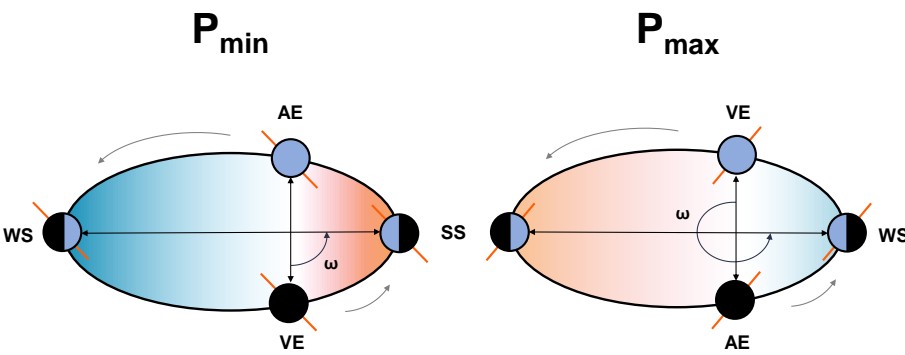

**Figure 1.** The schematic diagram showing the orbital configuration of minimum precession ($P_{min}$) and maximum precession ($P_{max}$). In $P_{min}$, summer solstice (SS) occurs at perihelion, while in $P_{max}$, winter solstice (WS) coincides with the perihelion. AE and VE are the Autumn and Vernal equinoxes respectively.

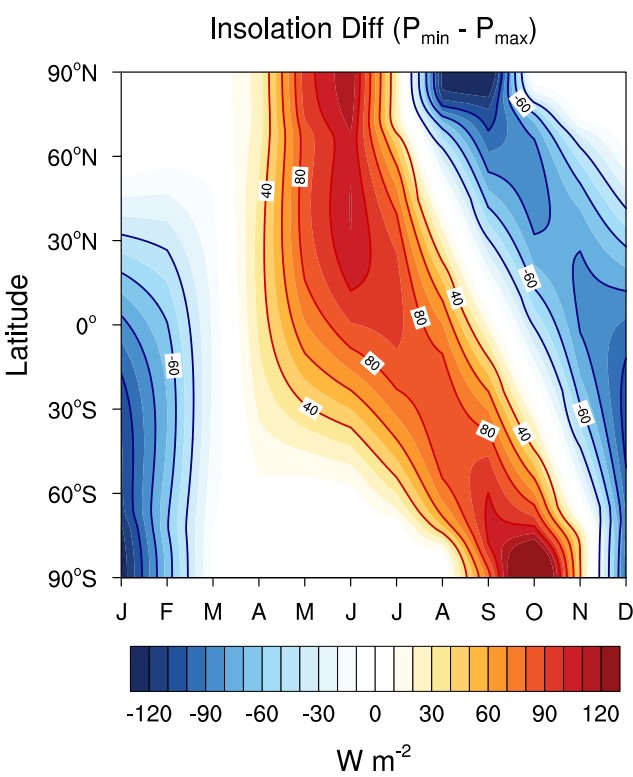

**Figure 2.** The difference in the incoming solar radiation at the top of atmosphere between $P_{min}$ and $P_{max}$ as a function of latitude and month.

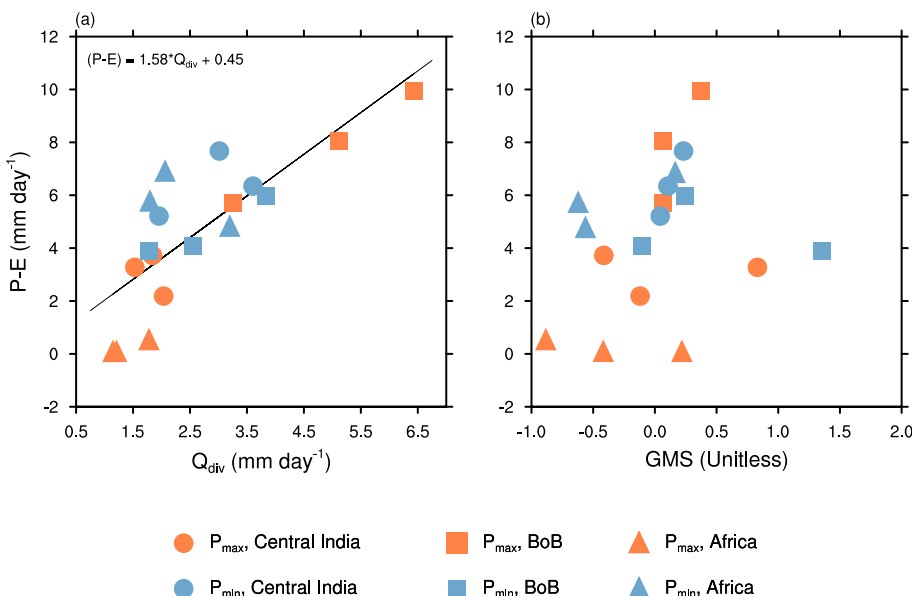

**Figure 3.** The dependence of $P - E$ on, **(a)** $Q_{div}$ and **(b)** GMS, for three regions: Central India (15°N - 25°N; 73°E - 83°E), the Bay of Bengal (10°N - 20°N; 85°E - 95°E) and Africa (5°N - 15°N; 20°W - 0E°). The months Jun-Jul-Aug are taken separately. The blue and orange symbols correspond to $P_{min}$ and $P_{max}$ respectively.

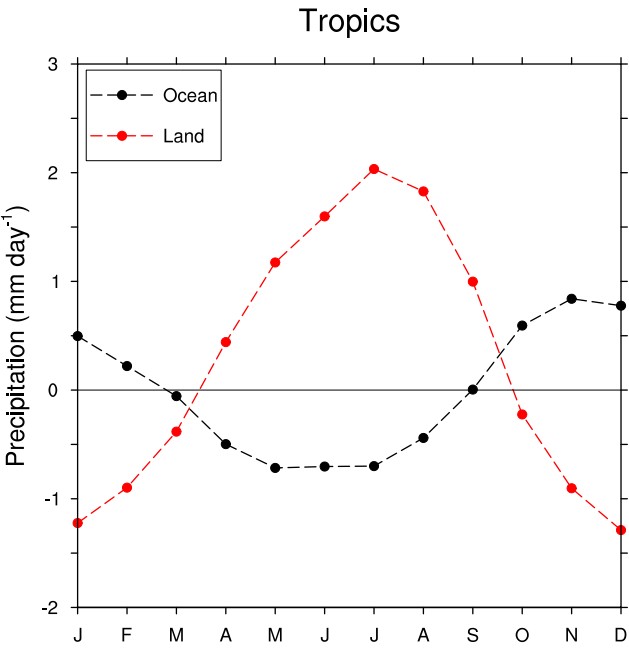

**Figure 4.** The difference in precipitation ($P_{min}$-$P_{max}$) for all the tropical land and ocean taken separately ($30°$S - $30°$N).

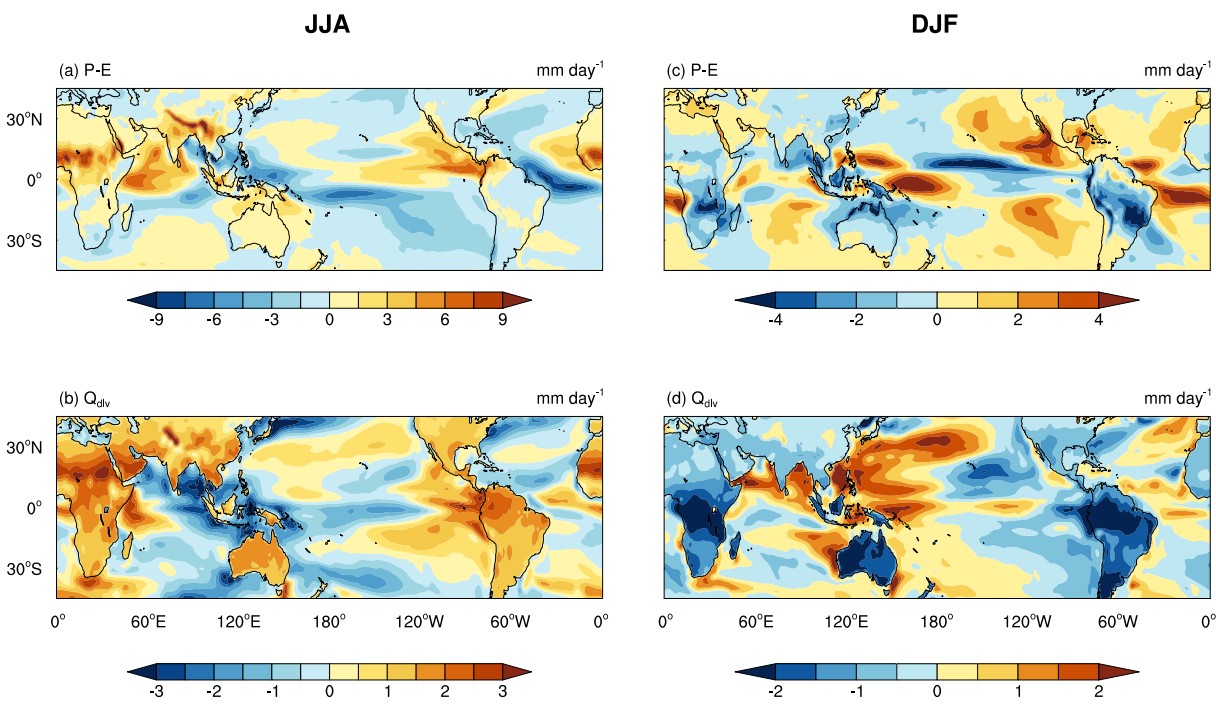

**Figure 5.** The difference in $P - E$ (top panel **(a)** and **(c)**) and $Q_{div}$ (bottom panel **(b)** and **(d)**). The left panel is for JJA mean and the right panel for DJF mean.

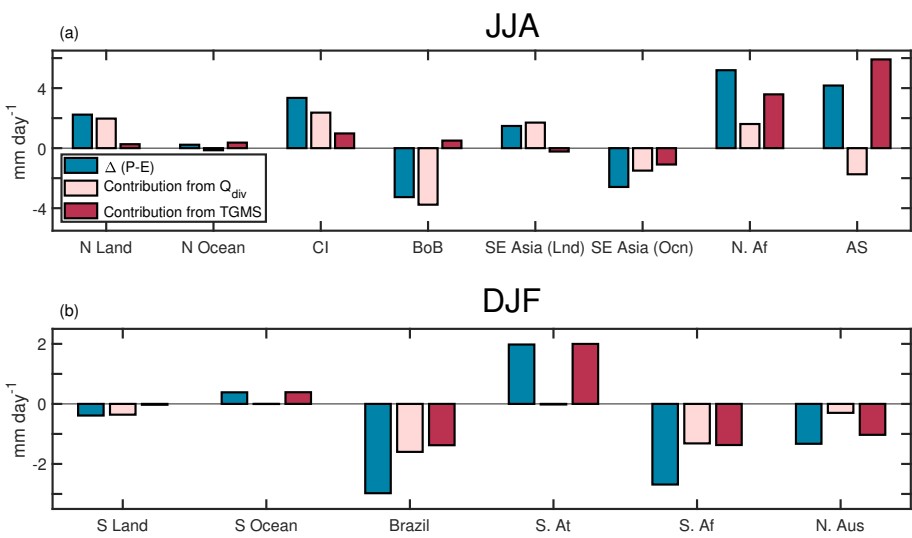

**Figure 6.** The contribution of $Q_{div}$ and TGMS to the changes in $P - E$. The top panel **(a)** is for the JJA mean and regions in the Northern Hemisphere, while the bottom panel **(b)** is for regions in the Southern Hemisphere and averaged over DJF. The blue bar is the change in $P - E$, while pink and red bars represent the contribution from $Q_{div}$ and TGMS. The abbreviations used in **(a)**, **N Land**: Northern tropics (land only), **N Ocean**: Northern tropics (Ocean only), **CI**: Central India, **BoB**: the Bay of Bengal, **SE Asia (Lnd)**: South East Asia (land only), **SE Asia (Ocn)**: South East Asia (Ocean only), **N. Af**: North Africa and **AS**: Arabian Sea, and in **(b)**, **S Land**: Southern tropics (land only), **S Ocean**: Southern tropics (ocean only), **S. At**: South Atlantic, **S. Af**: South Africa, **N. Aus**: North Australia. The coordinates of these regions are provided in Table 2.

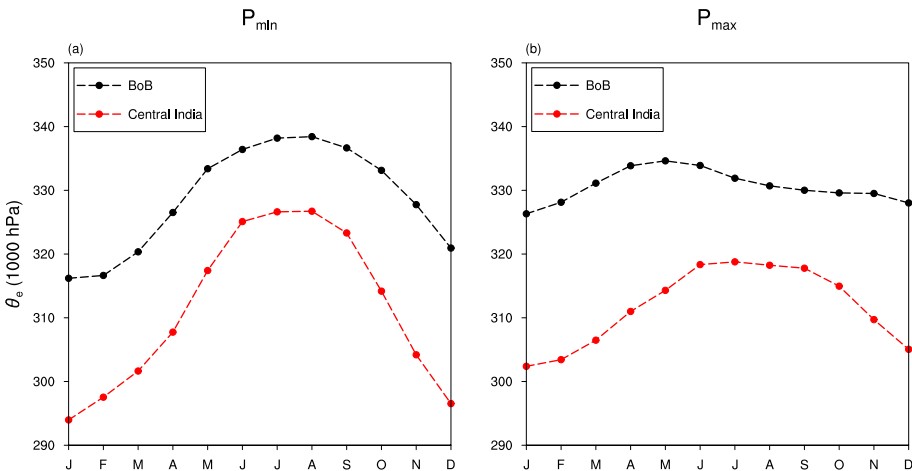

**Figure 7.** The seasonal cycle of near-surface equivalent potential temperature ($\theta_e$) over for India and the Bay of Bengal for, **(a)** $P_{min}$ configuration, and **(b)** $P_{max}$

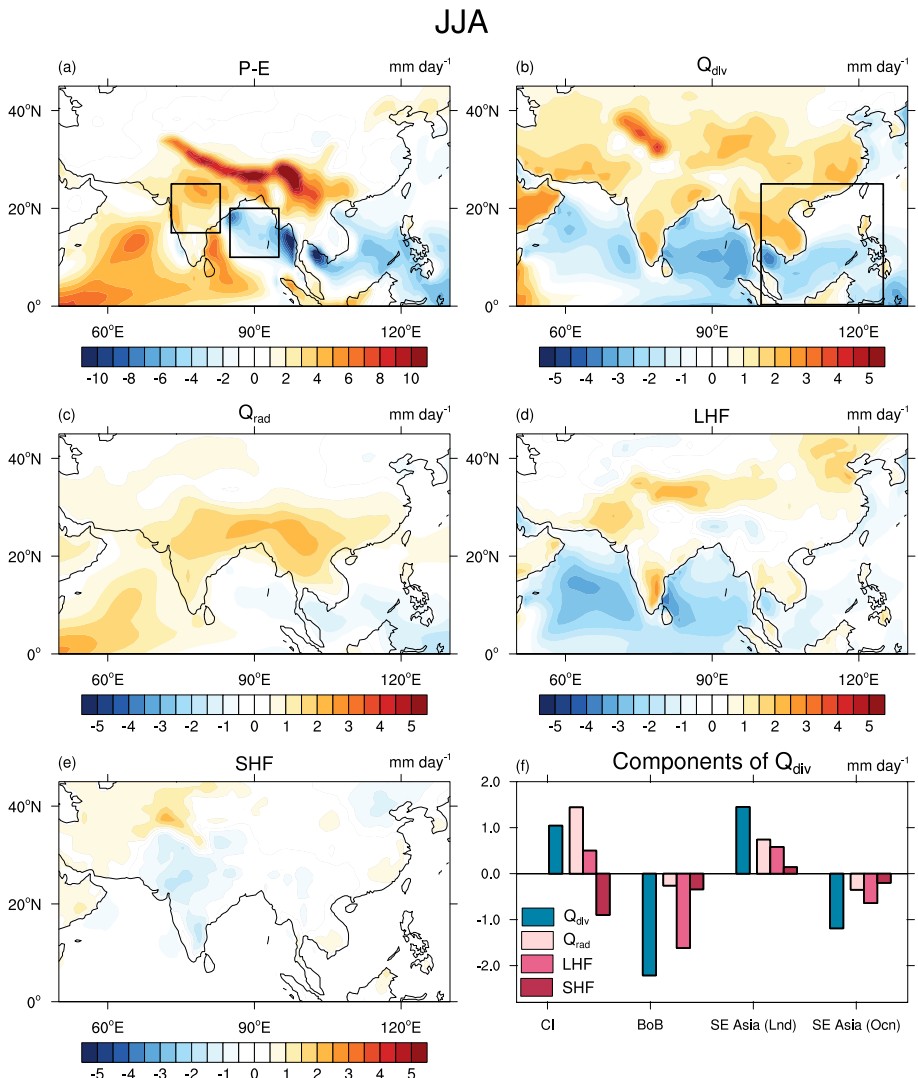

**Figure 8.** The JJA mean difference (P$_{min}$-P$_{max}$), in **(a)** $P - E$, **(b)** $Q_{div}$ (sum of energuy fluxes at the top and bottom of the atmosphere), **(c)** $Q_{rad}$ (Sum of all radiative fluxes at the top and bottom of the atmosphere), **(d)** Latent heat flux, **(e)** Sensible heat flux. The boxes shown in **(a)** and **(b)**, are the regions chosen for this study: Central India (15°N-25°N; 73°E-83°E), the Bay of Bengal (10°N-20°N; 85°E-95°E) and South East Asia ( 0°N-25°N; 100°E- 125°E). **(f)** shows the decomposition of $Q_{div}$ into radiative, latent and sensible heat fluxes for the two regions.

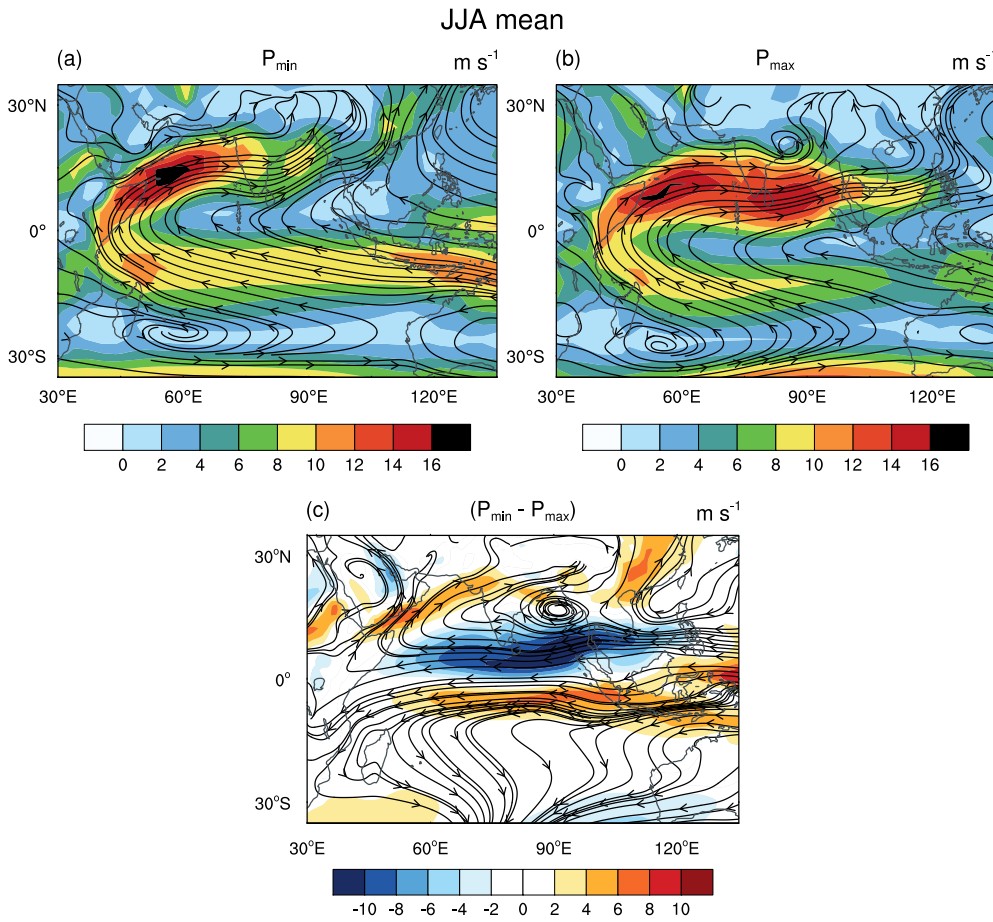

**Figure 9.** The JJA mean wind speed (850 hPa) in shading for **(a)** $P_{min}$, **(b)** $P_{max}$, with streamlines of the wind vector field superimposed. The difference of the winds between $P_{min}$ and $P_{max}$ is shown in **(c)**. 40°E longitude has a convergence at the equator and cyclonic circulation over the Middle-East. An anti-cyclonic circulation exists in the southern hemisphere over Madagascar. This is similar to the response of the atmosphere to equatorial plus off-equatorial heating (Gill, 1980).

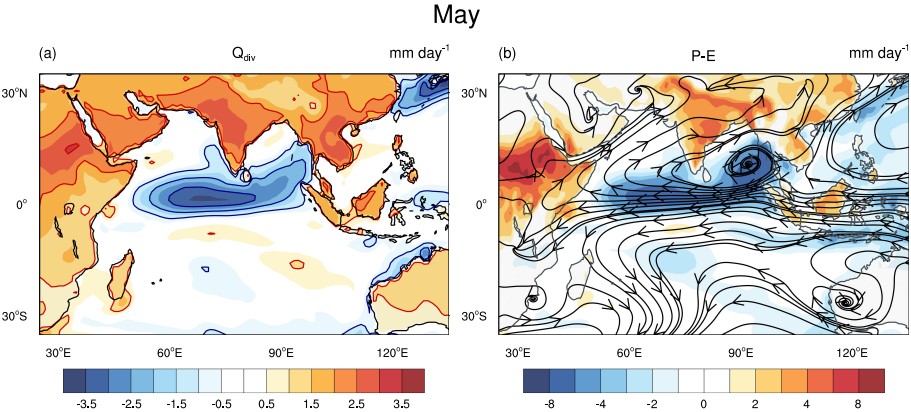

**Figure 10.** The difference between P<sub>min</sub> and P<sub>max</sub> in **(a)**, $Q_{div}$ and **(b)**, $P - E$ along with streamlines of change in the wind, for the month of May. Note that the large increase in $Q_{div}$ over Africa leads to an early onset of African monsoon. Thus, influencing the winds over the equatorial Indian Ocean.