# Peer review of "The response of tropical precipitation to Earth's precession: The role of energy fluxes and vertical stability"

_Climate of the Past, 2018_

## Referee Comment (RC1) · Anonymous Referee #1 · 6 Sep 2018

**Dear Dr. Yin,**

First thanks very much for assigning me to review this manuscript entitled "The response of tropical precipitation to Earth's precession: The role of fluxes and vertical stability" This work attempts to provide a dynamical explanation for comparing responses of tropical precipitation to $P_{min}$ and $P_{max}$. The topics is very interesting. However, much work needs to improve the quality of this work from the organization structures to Figures arrangement and conclusions. For this reason, I recommend to be published on the Journal of Climate of the Past after major revision.

**Comments on the manuscript**

The topic entitled "*The response of tropical precipitation to Earth's precession: The role of fluxes and vertical stability*" aims to provide a dynamical explanation for response differences of tropical precipitation to $P_{min}$ and $P_{max}$. Substantial works needs to do and major revisions are given. To help revision, I list the issues of my concerns.

1. Title: fluxes is too general, suggest "energy fluxes" replace fluxes

2. The structures of current manuscript needs to improve and arrange as follows:

First part is Introduction, and second part is Model description, and experiments design, methodology. The third part is results shows. Conclusions and discussion appear last part. However, the authors should move most of (maybe all) the formulas in section 2 with a quite clear description. Please re-organize this part.

3. Table 1 and Table 2 are not very beautiful.

4. All the title of Figurexx is not correct in your figure captions, because one more Arabic number "1" need to remove.

5. give a regression line for Figure 13 and Figure 14 and then merge into one plot.

6. All the Figures needs to make it more beautiful.

---

## Referee Comment (RC2) · Anonymous Referee #2 · 1 Nov 2018

**Comments on "The response of tropical precipitation to Earth's precession: The role of fluxes and vertical stability" by Jalihal et al.**

Effect of astronomical parameters on global monsoon and precipitation is one of the hottest topics, both for the proxy and modeling community. In this study, the authors tried to evaluate the precession effect on the tropical precipitation from a perspective of energy and moisture budget, with a special focus on the differences between land and ocean. It is a useful method and gives new understanding for precession effect. However, the paper in its present form seems a little preliminary and casual, especially its structure. I recommend the manuscript to be major revised before published in Climate of the Past. My comments are listed as follows,

(1) The structure of the text seems casual. I recommend that the authors put all the methods (including the ITCZ model, equation and decomposition) together. In the result section, it is better to merely show the figure and descriptions. That would help the paper to be easily read.

(2) Introduction: there are a lot of modeling studies in this field; however, the authors did not mention them in the introduction. For example, Global:

Kutzbach, J., Liu, X., Liu, Z., Chen, G., 2008. Simulation of the

evolutionary response of global summer monsoons to orbital forcing over the past 280,000 years. Clim. Dyn. 30, 567-579.

Asia and Africa:

Tuenter, E., Weber, S., Hilgen, F., Lourens, L., Ganopolski, A., 2005. Simulation of climate phase lags in response to precession and obliquity forcing and the role of vegetation. Clim. Dynam. 24, 279-295

Weber, S., Tuenter, E., 2011. The impact of varying ice sheets and greenhouse gases on the intensity and timing of boreal summer monsoons. Quat. Sci. Rev. 30, 469-479.

Shi, Z., Liu, X., Cheng, X., 2012. Anti-phased response of northern and southern East Asian summer precipitation to ENSO modulation of orbital forcing. Quat. Sci. Rev. 40, 30-38.

Caley, T., Roche, D.M., Renssen, H., 2014. Orbital Asian summer monsoon dynamics revealed using an isotope-enabled global climate model. Nat. Commun. 5, 5371.http://dx.doi.org/10.1038/ncomms6371.

Shi, Z., 2016. Response of Asian summer monsoon duration to orbital forcing under glacial and interglacial conditions: implication for precipitation variability in geological records. Quat. Sci. Rev. 139, 30-42

(3) Experiments: Only two sensitivity runs are conducted in this study. The authors said the differences between Pmax and Pmin scenarios "has a similar spatial precipitation response as observed in Mid

Holocene, but with higher amplitude". In actual, there is certain contribution from obliquity in the MH-PI difference. I know in Bosmans et al (2015), there are already obliquity-linked experiments. Why do the authors not give results for the obliquity in this study? In my opinion, it is also important.

(4) Results: From figure 5 and 6, I can see the distinct response of land and ocean precipitation, but it is also significantly negative over northwestern Pacific besides the Bay of Bengal. This indicates that the East Asian/Northwestern Pacific summer precipitation is also typical for the proposal of this paper. I recommend the authors to add additional analyses on this region and compare the results to those over the Bay of Bengal.

---

## Author Response (AR1)

**Author's response**

We thank the anonymous referees for their comments. We have addressed their concerns and have rewritten the manuscript. Our responses to the referees, a list of the changes made, and the marked-up version of the manuscript follows.

**Interactive comment on* "The response of tropical precipitation to Earth's precession: The role of fluxes and vertical stability", *reply to referee 1**

Chetankumar Jalihal[1,3], Joyce Helena Catharina Bosmans[2], Jayaraman Srinivasan[3], and Arindam Chakraborty[1,3]

[1]Centre for Atmospheric and Oceanic Sciences, Indian Institute of Science, Bangalore, India
[2]Department of Environmental Science, Radboud University, The Netherlands
[3]Divecha Centre for Climate Change, Indian Institute of Science, Bangalore, India

**Correspondence:** Chetankumar Jalihal (jalihal@iisc.ac.in)

1. **General Comments:** The authors thank you for your time and inputs.

2. ***Comment:*** *Title: fluxes is too general, suggest "energy fluxes" replace fluxes*

   ***Reply:*** Thank you for the suggestion. We will modify the title to include "energy fluxes".

3. ***Comment:*** *The structures of current manuscript needs to improve and arrange as follows: First part is Introduction, and second part is Model description, and experiments design, methodology. The third part is results shows. Conclusions and discussion appear last part. However, the authors should move most of (maybe all) the formulas in section 2 with a quite clear description. Please re-organize this part.*

10   ***Reply:*** We will move all the equations to section 2, and explain the equations in more detail.

4. ***Comment:*** *Table 1 and Table 2 are not very beautiful.*

   ***Reply:*** We will make better tables for the revised manuscript.

15   5. ***Comment:*** *All the title of Figurexx is not correct in your figure captions, because one more Arabic number "1" need to remove.*

   ***Reply:*** This was due to a LaTeX error, which had been fixed in the uploaded version of the discussion paper.

6. ***Comment:*** *give a regression line for Figure 13 and Figure 14 and then merge into one plot.*

20   ***Reply:*** Figure 3 and 4 in the discussion paper show the dependence of (P-E) on energy fluxes and stability of the

[Figure]

**Figure 1.** This figure shows the dependence of (P-E) on, **(a)** $Q_{div}$ (which is sum of all the energy fluxes into the atmosphere) and **(b)**, GMS. The scatter is for three regions: Central India (15$^o$N-25$^o$N; 73$^o$E-83$^o$E), Bay of Bengal (10$^o$N-20$^o$N; 85$^o$E-95$^o$E) and Africa (5$^o$N-15$^o$N; 20$^o$W-25E$^o$). The months JJA have been taken separately. (P-E) is directly proportional to $Q_{div}$.

atmosphere. Your point is well taken, and the two figures have been combined. (P-E) being a non-linear function of stability, the regression line is not being shown for the scatter of (P-E) and GMS.

7. ***Comment****: All the Figures needs to make it more beautiful.*

5    ***Reply****:*    We will improve all the figures for the revised manuscript.

**Interactive comment on* "The response of tropical precipitation to Earth's precession: The role of fluxes and vertical stability", *reply to referee 2**

Chetankumar Jalihal[1,3], Joyce Helena Catharina Bosmans[2], Jayaraman Srinivasan[3], and Arindam Chakraborty[1,3]

[1]Centre for Atmospheric and Oceanic Sciences, Indian Institute of Science, Bangalore, India
[2]Department of Environmental Science, Radboud University, The Netherlands
[3]Divecha Centre for Climate Change, Indian Institute of Science, Bangalore, India

**Correspondence:** Chetankumar Jalihal (jalihal@iisc.ac.in)

1. **General Comments:** The authors thank you for your time and inputs.

2. *Comment: The structure of the text seems casual. I recommend that the authors put all the methods (including the ITCZ model, equation and decomposition) together. In the result section, it is better to merely show the figure and descriptions.*
5          *That would help the paper to be easily read.*

   *Reply*:         Thank you for the suggestion. We will re-organize the manuscript.

3. *Comment: Introduction: there are a lot of modeling studies in this field; however, the authors did not mention them in the introduction. For example,*

10    *Global:*

   *Kutzbach, J., Liu, X., Liu, Z., Chen, G., 2008. Simulation of theevolutionary response of global summer monsoons to orbital forcing over the past 280,000 years. Clim. Dyn. 30, 567-579.*

   *Asia and Africa:*

15    *Tuenter, E., Weber, S., Hilgen, F., Lourens, L., Ganopolski, A., 2005. Simulation of climate phase lags in response to precession and obliquity forcing and the role of vegetation. Clim. Dynam. 24, 279-295*

   *Weber, S., Tuenter, E., 2011. The impact of varying ice sheets and greenhouse gases on the intensity and timing of boreal summer monsoons. Quat. Sci. Rev. 30, 469-479.*

   *Shi, Z., Liu, X., Cheng, X., 2012. Anti-phased response of northern and southern East Asian summer precipitation to ENSO modulation of orbital forcing. Quat. Sci. Rev. 40, 30-38.*

*Caley, T., Roche, D.M., Renssen, H., 2014. Orbital Asian summer monsoon dynamics revealed using an isotope-enabled global climate model. Nat. Commun. 5, 5371.http://dx.doi.org/10.1038/ncomms6371.*

*Shi, Z., 2016. Response of Asian summer monsoon duration to orbital forcing under glacial and interglacial conditions: implication for precipitation variability in geological records. Quat. Sci. Rev. 139, 30-42*

5      *Reply***:**      Thank you for these references. We will include them in the introduction.

4.   *Comment:   Experiments: Only two sensitivity runs are conducted in this study. The authors said the differences be-tween Pmax and Pmin scenarios "has a similar spatial precipitation response as observed in MidHolocene, but with higher amplitude". In actual, there is certain contribution from obliquity in the MH-PI difference. I know in Bosmans*
10      *et al (2015), there are already obliquity-linked experiments. Why do the authors not give results for the obliquity in this study? In my opinion, it is also important.*

*Reply***:**      It is true that the Mid-Holocene response will have contributions from obliquity. However, the obliquity in Mid-Holocene is only 0.66º higher than PI. Hence, its contribution is much less in comparison to that of precession.

[Figure]

**Figure 1.** This figure shows that the contribution from tilt to the total insolation change is much smaller in comparison to that from precession. The difference in insolation between, **a)** MH and PI, **b)** MH (obliquity only) and PI.

15      Our analysis is still valid for MH. It is discussed in detail in the section **Results and Discussion-1.1** of this document. The asymmetric precipitation response over India and Bay of Bengal is driven by the same mechanism, in MH and $P_{min}$.

Obliquity has a much smaller forcing than precession. The land-ocean asymmetric response in tropical precipitation still exists, albeit much weaker at the regional scale. The mechanisms governing these changes are different and hence were not included in the manuscript. **Results and Discussion section 1.2** here, has the relevant discussion.

5. ***Comment***: *Results: From figure 5 and 6, I can see the distinct response of land and ocean precipitation, but it is also significantly negative over northwestern Pacific besides the Bay of Bengal. This indicates that the East Asian/Northwestern Pacific summer precipitation is also typical for the proposal of this paper. I recommend the authors to add additional analyses on this region and compare the results to those over the Bay of Bengal.*

***Reply***: The response of East Asian/Northwestern Pacific precipitation is shown in the following figure. It will be included in the revised manuscript.

[Figure]

**Figure 2.** For the extreme precession experiments (.viz. $P_{min}$ and $P_{max}$), the bar chart shows, **a)** the relative contribution of $Q_{div}$ and TGMS to the changes in (P-E) and, **b)** the changes in the various fluxes contributing to $Q_{div}$. Here $Q_{div}$ is the sum of all the fluxes at the surface and top of atmosphere, $Q_{rad}$ is the sum of all the radiation fluxes (top and bottom of atmosphere), LHF and SHF are the latent and sensible heat fluxes. ($Q_{div} = Q_{rad} + LHF + SHF$). **CI:** Central India (15$^o$, 25$^o$N; 73$^o$, 83$^o$E) **BoB:** Bay of Bengal (10$^o$, 20$^o$N; 85$^o$, 95$^o$E) **SE Asia:** South East Asia (0$^o$, 25$^o$N; 100$^o$, 125$^o$E)

Figure (2a), shows that the dominant reason for the changes in (P-E) between $P_{min}$ and $P_{max}$, is $Q_{div}$ for all the four regions chosen. The land regions (Central India and South East Asia) show an increase in $Q_{div}$ (and hence in (P-E)), while over the oceanic regions (BoB and NW Pacific) $Q_{div}$ (and hence (P-E)) decreases. The increased insolation drives the positive changes in $Q_{div}$ over land regions, whereas the decrease in LHF is the main cause of decrease in $Q_{div}$ over

oceanic regions (Figure 2b). This decrease in the surface latent heat fluxes over the North-West Pacific is due to the reduction in wind speeds. This in turn, is a response to the convective heating of atmosphere over the West Equatorial Indian ocean and the Red Sea. This is the same reason why wind speed over the Bay of Bengal reduces (discussed in detail in the discussion paper).

**5  1   Results and Discussions**

**1.1   ($P_{min}$-$P_{max}$) vs (MH-PI):**

In this section, we have discussed the similarities between the sets of experiments ($P_{min}$, $P_{max}$) and (MH, PI). All four of these experiments used the model EC-Earth. The details of the experiments can be found in (Bosmans et al., 2012, 2015). Both the sets of simulations exhibit a land-ocean asymmetry in the response of precipitation to orbital foricngs. The amplitude of the 10   response is nearly a third in the (MH-PI) in comparison to that of ($P_{min}$-$P_{max}$) (Figure 3). Figure 4 shows the spatial response of (P-E) and $Q_{div}$. For MH, the response is quite similar in pattern to $P_{min}$, but of a smaller amplitude. There are some differences at the regional scale. This could be due to the effect of obliquity. However, the analysis that we have used in the discussion paper can still be used. The India-Bay of Bengal land-ocean asymmetry is due to the same mechanism in both MH and $P_{min}$.

[Figure]

**Figure 3.** Seasonal cycle of the change in precipitation over all the tropical land and ocean taken separately, for **a)** ($P_{min}$-$P_{max}$) and, **b)** (MH-PI).

[Figure]

**Figure 4.** Spatial patterns of the changes in (P-E) (left panel) and $Q_{div}$ (right panel), averaged over JJA. The top panel is for ($P_{min}$-$P_{max}$) and the bottom panel is for (MH-PI). There is a remarkable spatial coherence, but with a different magnitude.

**1.2 Obliquity experiments:**

In this section, we have discussed the response of tropical precipitation to obliquity forcing. For these experiments eccentricity was set to zero (circular orbit). The maximum and minimum obliquity experiments ($T_{max}$ and $T_{min}$) have a tilt of 24.45° and 22.08°, respectively. Further details of the experiments can be found in (Bosmans et al., 2015). Figure 5a shows the difference in insolation between the two obliquity experiments. The obliquity forcing is much smaller than the precessional forcing. However, there still exists a land-ocean asymmetry in the response of tropical precipitation, though of much smaller magnitude (Figure 5b).

[Figure]

(Tmax - Tmin)

[Figure]

-40  -30  -20  -10   0   10   20   30   40

W/m²

(a) Difference in insolation for the two obliquity experiments, $T_{max}$ and $T_{min}$.

[Figure]

Seasonal Cycle; (Tmax-Tmin)

(b) Seasonal cycle of the change in precipitation over all of tropical land and ocean taken separately.

**Figure 5.** Figures showing the obliquity forcing and the precipitation response.

The spatial patterns of the response of (P-E) are in general similar to that of precession (Figure 6). There are however, some regional differences. Using equation 13 of the discussion paper to identify the cause of these changes, reveals a much different mechanism than that for precession. For example, over Central India the changes in (P-E) is due to $Q_{div}$ in the precession experiments (Figure 2a) and due to TGMS in the obliquity experiments (Figure 7a). Even though over BoB, $Q_{div}$ causes a decrease in (P-E) for both precession and obliquity experiments, $Q_{div}$ decreases for different reasons. Precessional forcing causes winds to decrease, which reduces latent heat fluxes. Winds (and hence latent heat flux) also decrease in the obliquity experiments, but the decrease is not large enough. Hence, the changes in net radiation fluxes ($Q_{rad}$) become equally important (Figure 8f).

$(T_{max} - T_{min})$

[Figure]

**Figure 6.** Spatial variation in (P-E) (top panel), and $Q_{div}$ (bottom panel), averaged over the months JJA (left panel) and DJF (right panel).

[Figure]

**Figure 7.** The bar chart shows the relative contribution of $Q_{div}$ and TGMS to the changes in (P-E) for the, **(a)** northern summer, and **(b)** southern summer. **N. Land:** Northern Tropics (land only) ( $0^oN$-$30^oN$; $0^oE$-$360^oE$); **N. Ocean:** Northern Tropics (ocean only); **CI:** Central India ($15^oN$-$25^oN$; $73^oE$-$83^oE$); **BoB:** Bay of Bengal ($10^oN$-$20^oN$; $85^oE$-$95^oE$); **AS:** Arabian Sea ( $10^oN$-$20^oN$; $60^oE$-$70^oE$); **N. Af:** North Africa ( $5^oN$-$15^oN$; $20^oW$-$0^oE$); **S. Land:** Southern Tropics (Land only) ($30^oS$-$0^oN$; $0^oE$-$360^oE$); **S. Ocean:** Southern Tropics (Ocean only) ($30^oS$-$0^oN$; $0^oE$-$360^oE$); **Brazil:** ($20^oS$-$10^oS$; $70^oW$-$50^oW$); **S. At:** South Atlantic ($20^oS$-$10^oS$; $30^oW$-$0^oE$); **S. Af:** South Africa ($20^oS$-$10^oS$; $15^oE$-$35^oE$); **N. Aus:** North Australia ($25^oS$-$15^oS$; $130^oE$-$140^oE$)

[Figure]

**Figure 8.** JJA mean difference between $P_{min}$ and $P_{max}$ for, **a)** (P-E), **b)** $Q_{div}$, **c)** $Q_{rad}$ (sum of all radiation fluxes), **d)** Latent Heat Fluxes, **e)** Sensible Heat Fluxes, **f)** the components of $Q_{div}$.

[revised manuscript text omitted]
 ($15^{\circ}$N-$25^{\circ}$N; $73^{\circ}$E-$83^{\circ}$E), Bay of Bengal ($10^{\circ}$N-$20^{\circ}$N; $85^{\circ}$E-$95^{\circ}$E) and South East Asia ( $0^{\circ}$N-$25^{\circ}$N; $100^{\circ}$E- $125^{\circ}$E). **(f)** shows the decomposition of $_{div}$$Q_{div}$ into radiative, latent and sensible heat fluxes for the two regions. $_{div}$$_{rad}$

[Figure]

**Figure 9.**  The JJA mean  wind speed (850 hPa) in shading  for **(a)** $P_{min}$ , **(b)** $P_{max}$   with streamlines of the wind vector field superimposed. The difference  of the winds  between $P_{min}$ and $P_{max}$ is shown in **(c)**. 40°E longitude has a convergence at the equator and cyclonic circulation over the Middle-East. An anti-cyclonic circulation exists in the southern hemisphere over Madagascar. This is similar to the response of  the atmosphere to equatorial plus off-equatorial heating (Gill, 1980).

[Figure]

**Figure 10.** The difference between $P_{min}$ and $P_{max}$ in **(a)**,  $Q_{div}$ and **(b)**,  $P-E$ along with  streamlines of change in the wind, for the month of May. This shows that  the large increase in  $Q_{div}$ over Africa  leads to an early onset of African monsoon. Thus, influencing the winds over the equatorial Indian Ocean.

The plot shows the seasonal cycle of near surface equivalent potential temperature for India and BoB in the, **(a)** $P_{min}$ configuration, and **(b)** $P_{max}$

---

## Author Response (AR2)

**Author's response**

We thank the anonymous referee #1 for the comments. We have addressed the concerns and have rewritten the manuscript. Our responses to the referees, a list of the changes made, and the marked-up version of the manuscript follows.

**Response to review* "The response of tropical precipitation to Earth's precession: The role of fluxes and vertical stability", *reply to referee 1**

Chetankumar Jalihal[1,3], Joyce Helena Catharina Bosmans[2], Jayaraman Srinivasan[3], and Arindam Chakraborty[1,3]

[1]Centre for Atmospheric and Oceanic Sciences, Indian Institute of Science, Bangalore, India
[2]Department of Environmental Science, Radboud University, The Netherlands
[3]Divecha Centre for Climate Change, Indian Institute of Science, Bangalore, India

**Correspondence:** Chetankumar Jalihal (jalihal@iisc.ac.in)

1. **General Comments:** The authors thank you for your time and inputs.

2. *Comment*: *structures of this paper after Introduction need to improve: For the title of Section 2 you can use words like this "Model and experimental description, methodology", subtitle of this Section 2: Section 2.1 Model and experiment*
5 *design; Section 2.2 diagnose methodology. This is only a suggestion.*
   *Reply*: Thank you for the suggestion. We have moved the current section 3 into section 2 as a subsection.

3. *Comment*: *The result description seems to begin since 5Page Line20 (i.e., Section 3 Results) The present title of subtitle, such as "4.1 Tropics"; "4.1.1 Qualitative analysis", "4.1.2 Quantitative analysis", are too general. Please revise them*
10 *by attractive titles.*
   *Reply*: We have improved these subtitles. All the results are in the "Results "section. The purpose of line 20 on page 5, is to provide justification for modifying the simple model for ITCZ. This is mentioned in lines 21 and 22 of pages 5 and 6: *We chose these three regions to highlight that neglecting the role of horizontal advection, may not always be appropriate.* After establishing the importance of using advection terms in the simple model, we have used the simple
15 model to diagnose precipitation response in various regions starting with the tropics and then the South Asian monsoon.

4. *Comment*: *For the results organization, I suggest the authors from the global average to the regional part. When you organize your work, please keep mind that the topic and its connection to the methodology you use to demonstrate or persuade the audiences or readers.*
20 *Reply*: The results were in the same order as suggested by the referee. We start with the analysis of the tropics as a whole and then talk about the Indian monsoon and the East Asian monsoon in detail.

5. **Comment**: *When you finish new revision, I am sure you need do some changes for this abstract and conclusions. For the second sentence in the current abstract, it is very easy to misunderstand. I think the authors want to express "This has been attributed to the changes in seasonal solar radiation at the top of the atmosphere"*

   **Reply**: We have rephrased the second sentence of the abstract.

6. **Comment**: *Please avoid the concluded sentences in the Figure description (Such Figure 4). I guess the author want to express "contrast responses of land ocean precipitation to different precession forcings (Pmin-Pmax)". Please Check all the Figure descrpitoins.*

   **Reply**: We have modified figure descriptions.

7. **Comment**: *For the introduction, "The paper is organized as follows" should begin from new paragraph.*

   **Reply**: The description of how the paper is organized now starts from a new paragraph.

8. **Comment**: *Recent monsoon dynamics based on MSE also need involve the reference list*
   *Sun Y., T. Zhou, G. Ramstein, C. Contoux and Z. Zhang, 2016: Drivers and mechanisms for enhanced summer monsoon precipitation over East Asia during the mid-Pliocene in the IPSL-CM5A. Climate Dynamics, 46,1437-1457 DOI: 10.1007/s00382-015-2656-4*
   *Sun Y, Ramstein Gilles, Li Laurent. Z X, Contoux Camille, Tan Ning, and Zhou Tianjun, 2018: Quantifying East Asian summer monsoon dynamics in the ECP4.5 scenario with reference to the mid-Piacenzian warm period. Geophysical Research Letters, 45,523-533,https://doi.org/10.1029/2018GL080061*

   **Reply**: We have included these references in the "Methodology" section.

**List of relevant changes made:**

1. Changed the second line of abstract, as per the referee's suggestions.

2. Subtitles have been modified.

3. Removed concluding sentences from the figure description.

4. Description of paper organization moved to a new paragraph.

5. References suggested by the referee included in the introduction, along with a description of the advantages of each method.

6. Moved the section "A simple model for ITCZ" to section 2 as a subsection with a title: "Diagnostic Methodology".

7. Split the section "Discussions and conclusions" into two sections: "Discussion" and "Summary and conclusions".

[revised manuscript text omitted]